# Gut Microbiota Induced by Pterostilbene and Resveratrol in High-Fat-High-Fructose Fed Rats: Putative Role in Steatohepatitis Onset

**DOI:** 10.3390/nu13051738

**Published:** 2021-05-20

**Authors:** Iñaki Milton-Laskibar, Laura Judith Marcos-Zambrano, Saioa Gómez-Zorita, Alfredo Fernández-Quintela, Enrique Carrillo de Santa Pau, J. Alfredo Martínez, María P. Portillo

**Affiliations:** 1Precision Nutrition and Cardiometabolic Health Program, IMDEA-Food Institute (Madrid Institute for Advanced Studies), Campus of International Excellence (CEI) UAM+CSIC, Spanish National Research Council, 28049 Madrid, Spain; inaki.milton@imdea.org (I.M.-L.); josealfredo.martinez@imdea.org (J.A.M.); 2CIBERobn Physiopathology of Obesity and Nutrition, Institute of Health Carlos III (ISCIII), 28029 Madrid, Spain; mariapuy.portillo@ehu.eus; 3Computational Biology Group, Precision Nutrition and Cancer Research Program, IMDEA Food Institute, 28049 Madrid, Spain; judith.marcos@imdea.org (L.J.M.-Z.); enrique.carrillo@imdea.org (E.C.d.S.P.); 4Nutrition and Obesity Group, Department of Nutrition and Food Science, Faculty of Pharmacy and Lucio Lascaray Research Centre, University of the Basque Country (UPV/EHU), 01006 Vitoria-Gasteiz, Spain; alfredo.fernandez@ehu.eus; 5BIOARABA Health Research Institute, 01006 Vitoria-Gasteiz, Spain

**Keywords:** (poly)phenols, resveratrol, pterostilbene, microbiota, fatty liver, steatohepatitis, rat, high-fat-high-fructose diet

## Abstract

Resveratrol and its 2-methoxy derivative pterostilbene are two phenolic compounds that occur in foodstuffs and feature hepato-protective effects. This study is devoted to analysing and comparing the metabolic effects of pterostilbene and resveratrol on gut microbiota composition in rats displaying NAFLD induced by a diet rich in saturated fat and fructose. The associations among changes induced by both phenolic compounds in liver status and those induced in gut microbiota composition were also analysed. For this purpose, fifty Wistar rats were distributed in five experimental groups: a group of animals fed a standard diet (CC group) and four additional groups fed a high-fat high-fructose diet alone (HFHF group) or supplemented with 15 or 30 mg/kg bw/d of pterostilbene (PT15 and PT30 groups, respectively) or 30 mg/kg bw/d of resveratrol (RSV30 group). The dramatic changes induced by high-fat high-fructose feeding in the gut microbiota were poorly ameliorated by pterostilbene or resveratrol. These results suggest that the specific changes in microbiota composition induced by pterostilbene (increased abundances of *Akkermansia* and *Erysipelatoclostridium*, and lowered abundance of *Clostridum* sensu stricto 1) may not entirely explain the putative preventive effects on steatohepatitis.

## 1. Introduction

Liver diseases have become a major health problem in the last few decades worldwide, accounting for approximately 2 million deaths per year [1]. Among such morbid conditions, non-alcoholic fatty liver disease (NAFLD) is one of the most common, which is usually associated with obesity [2]. NAFLD encompasses several stages, hepatic steatosis being the most benign [3]. This hepatic injury features an intrahepatic triglyceride (TG) accumulation greater than 5% of the total liver weight [4].

One of the causes leading to hepatic steatosis is excessive fructose consumption [5], which has been reported to promote lipid deposition in the liver by stimulating *de novo* lipogenesis in this organ [6]. In addition, fructose can contribute to the progression of hepatic steatosis towards more harmful stages of liver damage by inducing oxidative stress and mitochondrial dysfunction [7]. All these events are well-known contributors to liver inflammation, which is a major process in hepatic damage progression [8].

Besides specific dietary components, much attention has been paid to gut function and microbiota composition as potential agents involved in the progression of NAFLD [9]. In this regard, impaired gut microbiota composition, accompanied by increased gut permeability, has been described in rats fed with high-fructose diets [10]. Similarly, it has been reported that subjects suffering from NAFLD present gut microbiota dysbiosis, although no specific microbiota composition pattern has been described for these patients so far [10].

In this context, no specific treatment has been designed for hepatic steatosis, the prescription of hypocaloric diets being the main therapeutic choice based on the fact that steatosis is commonly found in obese people. Although the effectiveness of hypocaloric treatments in the amelioration of hepatic steatosis has been studied [4,11,12,13], the low adherence of patients to this protocol, especially in the long term, constitutes one of the main limitations of this nutritional strategy.

In this scenario, bioactive compounds have gained interest as a potential therapeutic approach for hepatic steatosis treatment. Among them, much attention has been paid to phenolic compounds, a wide and heterogeneous group of molecules that naturally occur in a variety of foods and food-derived products commonly included in our diet. One such compound is resveratrol (3,5,4′-trihydroxy-*trans*-stilbene), which has been extensively investigated due to its known capacity to mimic caloric restriction [4,11,14]. Indeed, resveratrol has shown to be effective in reducing hepatic lipid content in rodents [15]. The results are not so clear in humans but there is evidence that resveratrol can improve some inflammatory markers [16]. Nevertheless, one of the main limitations when considering resveratrol as a therapeutic agent is its low bioavailability, mainly due to the extensive metabolism that it undergoes once it reaches the gut and the liver [17,18]. In this context, pterostilbene has emerged as an alternative to resveratrol, since this resveratrol derivative is characterised by the presence of a single hydroxyl group in the chemical structure, which confers greater metabolic stability compared to resveratrol [19]. Studies addressing the effects of pterostilbene in hepatic steatosis are so far scarce, but it has been reported that this phenolic compound partially prevents liver steatosis in high-fat high-fructose fed rats by modulating lipid metabolism (reducing *de novo* lipogenesis and enhancing TG assembly and release, and improving mitochondrial function [20,21]). In addition, it has also been reported that both resveratrol and pterostilbene can effectively modify gut microbiota composition and diversity in a variety of experimental rodent models (mice and rats) as well as in humans [22,23,24,25,26]. Thus, it cannot be disregarded that the benefits described for resveratrol and pterostilbene in hepatic steatosis may, at least partially, be attributed to the modulation by both phenolic compounds in the gut microbiota.

In this context, this study is devoted to analysing and comparing the effects of pterostilbene and resveratrol in gut microbiota composition in rats fed with a diet rich in saturated fat and fructose, which induces the development of NAFLD. In addition, the current study is aimed at examining associations among the changes induced by both phenolic compounds in liver lipid content and the variations induced in gut microbiota composition.

## 2. Materials and Methods

### 2.1. Animals, Diets and Experimental Design

The study was performed utilising 50 six-week-old male Wistar rats purchased from Envigo (Barcelona, Spain). The animals were housed in polycarbonate metabolic cages under controlled temperature conditions (22 °C) and subjected to a 12-h light/dark cycle. After a 6-day adaptation period, the rodents were randomly distributed into five groups of ten animals each: the control group was fed with a standard diet (AIN-93G, OpenSource Diets, Denmark, D10012G); rats in the HFHF group received a high-fat high-fructose diet (OpenSource Diets, Denmark, D09100301); rodents in the PT15 and PT30 groups were fed with a high-fat high-fructose diet and supplemented with 15 or 30 mg pterostilbene/kg body weight/day (respectively); and the animals in the RSV30 group were provided the same high-fat high-fructose diet supplemented with 30 mg resveratrol/kg body weight/day. Pterostilbene was kindly supplied by ChromaDex Inc. (Irvine, CA, USA) and resveratrol by Monteloeder (Elche, Alicante, Spain). Both phenolic compounds were incorporated into the powdered diets on a daily basis, as previously reported [27]. After 8 weeks under these experimental conditions, the animals were anesthetised (chloral hydrate) and sacrificed by cardiac exsanguination after fasting (12 h). Livers were dissected, weighed and immediately frozen in liquid nitrogen. A piece of the small lobule of the liver of each animal was separated for histologic analysis. Serum was obtained from blood samples after centrifugation (1000× *g* for 10 min, at 4 °C). All samples were stored at −80 °C until analysis.

Fresh faecal samples were collected directly after defecation at the end of the intervention period, prior to the overnight fasting. Each animal was taken one at a time and housed in a clean, single cage with the aim of obtaining faeces directly after defecation. A soft abdominal massage was also applied to the animals in order to facilitate bowel movement and, in turn, the collection of fresh faeces. Samples were gathered in Falcon tubes and immediately frozen at −80 °C for future analyses. 

All the experiments were performed in agreement with the Ethical Committee of the University of the Basque Country (document reference CUEID CEBA/30/2010), according to the European regulations (European Convention—Strasburg 1986, Directive 2003/65/EC and Recommendation 2007/526/EC).

### 2.2. Histopathological Evaluation of NAFLD

The histological analysis was performed by light microscopy immediately after sacrifice. For this purpose, liver samples were fixed in 10% buffered formalin and subsequently embedded in paraffin. Liver sections were stained with both haematoxylin and eosin along with Masson’s trichrome using standard techniques.

The evaluated features were as follows: steatosis, lobular inflammation, ballooning degeneration and fibrosis. Lobular inflammation was graded based on the number of inflammatory foci per 200 x field: 0 (None) without inflammatory foci; 1 (Mild) when 1 or 2 inflammatory foci were found; and 2 (Moderate) when 2 to 4 inflammatory foci were observed. Regarding ballooning, this parameter was evaluated based on the number of affected hepatocytes displayed in this lesion: 0, when no ballooning was observed, 1 when few hepatocytes showed ballooning, and 2 when many cells exhibited ballooning degeneration. Regarding the NAS score, it was calculated by means of data from steatosis, lobular inflammation, ballooning and fibrosis scores.

### 2.3. Fecal DNA Extraction and 16S rRNA Gene Amplification for Microbiota Composition Analysis

DNA extraction was performed using the QIAamp DNA stool MiniKit according to the manufacturer’s instructions (QIAGEN, Hilden, Germany). DNA quantity and integrity were checked through LabChip GX (PerkinElmer, Waltham, MA, USA).

For the metagenomic study, the variable V3 and V4 regions of the prokaryotic 16S ribosomal RNA gene (16S rRNA) were analysed. The full-length primer sequences are as follows:Forward: 5′ TCGTCGGCAGCGTCAGATGTGTATAAGAGACAGCCTACGGGNGGCWGCAG;Reverse: 5′-GTCTCGTGGGCTCGGAGATGTGTATAAGAGACAGGACTACHVGGGTATCTAATCC.

Amplicon preparation was performed by using the 16S Metagenomic Sequencing Library Preparation Protocol (Illumina, San Diego, CA, USA). This protocol also includes overhang adapter sequences for compatibility with Illumina index and sequencing adapters. Amplicon size was subsequently verified by electrophoresis (LabChip GX; PerkinElmer, Waltham, MA, USA).

DNA libraries for 16S rRNA amplicon sequencing were prepared using the Nextera XT DNA Library Preparation Kit (Nextera XT) (Illumina, San Diego, CA, USA) according to the manufacturer’s instructions. The 16S rRNA libraries were sequenced with the Illumina MiSeq benchtop sequencer in paired-end mode with 2 × 300 cycles using the MiSeq Reagent v3 600-cycle kit (Illumina, San Diego, CA, USA).

The low-quality reads were filtered and chimeric sequences were removed after alignment using the Quantitative Insights into Microbial Ecology program (QIIME2) [28]. The subsequent clean reads were clustered as amplicon sequence variants (ASVs) using DADA2 [29] and annotated with the SILVA v.132 16S rRNA gene database [30]. Relative abundance of each ASV and alpha-diversity was calculated using the Phyloseq R package [31]. Beta-diversity was estimated by calculating weighted UniFrac distances and then visualised by means of principal coordinate analysis (PCoA). ANOSIM was achieved to compare bacterial communities’ similarity among groups using the “vegan” package of R (version 2.5-7). Linear discriminant analysis (LDA) effect size (LEfSe) was calculated to identify the bacterial taxa differentially enriched in different bacterial communities [32]. 

### 2.4. Statistical Analysis

Results are presented as mean ± SEM. Statistical analysis was performed using SPSS 24.0 (SPSS, Chicago, IL, USA). Data were analysed by one-way ANOVA followed by Newman–Keuls *post-hoc* test. In the case of differences in the abundance of taxa, statistical analysis was performed using the Kruskal–Wallis test. Significance was assessed at the *p* < 0.05 level.

Correlations between differentially enriched bacterial taxa and the liver steatosis and inflammation parameters were estimated by Spearman rank method, using the Microbiome package in R (http://microbiome.github.io/microbiome/, accessed on 15 February 2021). The correlation was assessed as coefficient ≥0.2 and FDR ≤ 0.05.

## 3. Results

### 3.1. General Parameters

Body weight gain was lower in animals fed with the control diet when compared with the animals fed with the high-fat high-fructose diet (HFHF group), as shown in Table 1. Whereas pterostilbene did not induce changes in this variable (PT15 and PT30 groups), resveratrol treatment led to a lower value than that of the HFHF group, failing to reach the value of the control group. In the case of food efficiency, no significant differences were observed between rats fed with the control diet or the high-fat high-fructose or among rats fed the high-fat high-fructose diets supplemented or not with the phenolic compounds.

An increase in liver weight was found in the HFHF, PT15 and PT30 groups when compared to the CC group. In the case of the group supplemented with resveratrol, the value was similar to that observed in the control group. Transaminase levels were strongly increased by high-fat high-fructose feeding. This effect was partially blocked by the phenolic compounds.

### 3.2. Liver Steatosis and Inflammation

Histological analysis revealed that in the HFHF group, one rat developed grade 3 steatosis, seven rats grade 2 steatosis and one rat grade 1 steatosis (note that, in this group, steatosis assessment was performed in 9 animals). In the PT15 group, six rats displayed grade 1 steatosis and four showed grade 2. In the PT30 and RSV30 groups, nine rats exhibited grade 1 steatosis and one rat showed grade 2 (Figure 1 and Figure 2). Regarding inflammation, rats from the CC group showed neither inflammation nor ballooning degeneration (Figure 1B,C). In the HFHF group, while half the rats had mild lobular inflammation, the other half showed moderate inflammation. In this group, all the animals displayed significant ballooning degeneration. Both doses of pterostilbene and resveratrol reduced lobular inflammation but not ballooning degeneration. In the PT15 group, eight rats showed mild inflammation whereas only two exhibited moderate inflammation. In the PT30 group, two rats showed no lobular inflammation and eight of them displayed mild inflammation. Finally, in the RSV30 group, seven rats showed mild inflammation and three moderate. Fibrosis was only observed in one rat which belonged to the HFHF group. Regarding the NAFLD score, the values from the CC, HFHF, PT15, PT30 and RSV30 groups were 0, 5.4 ± 0.4, 3.8 ± 0.3, 3.1 ± 0.3 and 3.8 ± 0.3, respectively.

### 3.3. Microbiota Composition and Diversity

Alpha-diversity of the microbiome was estimated using the Chao1, Shannon and Simpson indexes. The samples from the control rats, fed with a standard diet, displayed higher diversity than the rats fed with the high-fat high-fructose diet, independent of the presence or not of phenolic compounds (Figure 3). Principal coordinates analysis (PCoA) based on Weighted UniFrac distance showed a separation in the gut microbiota structure between the rats fed with the control diet and those fed with the high-fat high-fructose diet, regardless of the ingestion of pterostilbene or resveratrol (Figure 4, Appendix A). The most abundant phyla present were *Firmicutes, Bacteroidetes, Verrucomicrobia, Actinobacteria* and *Proteobacteria*. Regarding genus level, differences were found according to the detailed treatment. Twenty discriminatory genera were identified with an abundance greater than 0.01% after performing the Kruskal–Wallis test among groups (Figure 5).

Results revealed that levels of *Ruminococcaceae* UCG-014, *Ruminococcaceae* UCG-005, *Ruminoclostridium* 9 and *Ruminococcus* 2 were higher in the control rats, the abundance of these bacteria being dramatically lower in the high-fat high-fructose fed animals. Indeed, in these animals, the aforementioned bacteria were replaced with those from the genus *Lactococcus*, UBA1819, *Clostridium* sensu stricto 1, *Blautia* and *Faecalitalea*. Slight differences in genus composition were noted in the rats treated with pterostilbene or resveratrol. In this regard, a greater abundance of *Akkermansia*, *Erysipelatoclostridium* and *Fourrnierella*, and lower levels of *Clostridium* sensu stricto 1, were found in the animals from the PT15 group, while the PT30 group showed higher levels of *Streptococcus*. In the case of the animals receiving resveratrol (RSV30 group), their microbiome was characterised by a greater presence of *Lactococcus,* and *Blautia*.

A LEfSe analysis was performed in order to find microbial biomarkers associated with each treatment. Thus, at the genus level, it was observed that the microbiota from PT15 rats was dominated by *Erysipelatoclostridium, Erysipelotrichia*, UBA1819, *Eubacterium coprostanoligenes* group and *Fournierella.* In the case of the rats from the PT30 group, their microbiota was enriched in *Streptococcus*, while that of the resveratrol treated rats was enriched in *Blautia*, *Moryella* and *Lactococcus* (Figure 6).

A correlation analysis seeking an association between microbial composition, on the one hand, and general parameters, liver histopathological parameters and serum parameters, on the other hand, was carried out using the twenty most abundant and differentially expressed genera. Seven genera were found to be associated with hepatic damage, characterised by a higher transaminase level, increased liver weight, steatosis, lobular inflammation, ballooning and a higher NAS score (*Lactococcus*, *Terrisporobacter, Blautia, Lachnoclostridium, Fournierella, Faecalitalea, Clostridium* sensu stricto 1). Additionally, nine genera were associated with a hepato-protective effect, determined by lower liver weight, transaminase levels, NAS score, steatosis, ballooning and lobular inflammation (*Ruminiclostridium* 9, *Ruminococcaceae* NK4A214 group, *Ruminococcaceae* UCG-005, *Eubacterium coprostanoligenes* group, *Firmicutes* bacterium CAG:822, *Streptococcus*, *Ruminococcus* 2, *Ruminococcaceae* UCG-014 and *Akkermansia*) (Appendix A).

## 4. Discussion

Fructose consumption has been on the rise in the last few decades [33]. Besides the consumer’s perception that fructose may be healthier than sucrose, the high availability of fructose-rich foods and beverages has contributed to this trend [33]. Interestingly, a high fructose consumption pattern seems to be concomitant with the increase in NAFLD prevalence [34].

The steatosis induced by the high-fat high-fructose diet was partially prevented by pterostilbene and resveratrol with similar effectiveness. Interestingly, pterostilbene not only showed a dose–response pattern in the hepato-protective effect, but it proved to be more effective than resveratrol in preventing the inflammation induced by the high-fat high-fructose diet. It could be hypothesised that this particular response could be attributable to the higher bioavailability of pterostilbene with regard to resveratrol, although a more specific anti-inflammatory activity caused by the differences in their chemical structure (the presence of two methoxy groups in pterostilbene instead of two hydroxyl groups) cannot be disregarded.

The “multiple hit” theory that is currently being used to describe the events involved in NAFLD development considers gut microbiota impairment as one of the potential etiological mechanisms [35]. It follows that the increase in pro-inflammatory cytokine release, which results from impaired intestinal barrier function, is one of the main causes of liver damage [35]. The administration of phenolic compounds such as resveratrol or pterostilbene has been proposed as an additional tool to improve the outcomes of the tactics used in the treatment of this liver condition [3,20,21]. However, the fact that the potential beneficial effects attributed to polyphenols in NAFLD may occur directly or as a result of the modulation exerted by these compounds in the gut microbiota still remains unclear [36].

The analysis of gut microbiota composition revealed that the animals fed with the high-fat high-fructose diet had a significantly reduced microbial α-diversity, which is in line with that described in previous studies using rodent models fed with diets high in saturated fat and/or sugar [37,38,39,40]. Interestingly, similar observations have also been made in humans following “westernised” diets [41,42]. Several reports show that the hepatic amelioration (lipid content and inflammation) induced by pterostilbene and resveratrol was not derived from an increased microbiota diversity but perhaps from the potential modulation of specific bacteria, as has been proposed elsewhere [23,43]. In this regard, the lowered abundance of the *Ruminococcaceae* family found in the high-fat high-fructose diet fed animals is in line with studies carried out in humans, where the abundance of these bacteria has been inversely associated with NAFLD [43]. Indeed, bacteria of the *Ruminococcaceae* family are considered important mediators of gut microbiome and gastrointestinal health in humans [44], since they improve gastrointestinal barrier integrity and, thus, they protect the liver [45,46]. Notably, the correlation analysis carried out in the present study highlighted the liver-protective effect of bacteria from various genera of the *Ruminococcaceae* family, since negative correlations were found between these genera and parameters such as liver weight, steatosis, ballooning and inflammation. The amelioration induced by pterostilbene and resveratrol in liver alterations was not due to changes in these genera, since no significant differences were observed in their levels after treatment with these phenolic compounds.

With regard to the groups fed with the high-fat high-fructose diet and treated with the tested compounds, some differences regarding the abundance of certain bacteria genera were found. Interestingly, a greater abundance of *Akkermansia* genus was identified in the animals treated with the low dose of pterostilbene (PT15 group), similar to previous observations carried out in our laboratory using a genetic model of obesity (*fa/fa* Zucker rat) treated with the same dose of pterostilbene (15 mg/kg bw/d) [23]. Taking into account that bacteria from this genus, such as *Akkermansia municiphila,* have been reported to enhance gut protection and to improve gut barrier function [23,47,48,49], the increased abundance of *Akkermansia* found in the PT15 group could, at least partially, be mediating the hepato-protective effect induced by the phenolic compound administration. In addition, the gut microbiota of the animals receiving the low pterostilbene dose was also enriched in *Erysipelatoclostridium*, known to promote tight junction proteins, which enhances intestinal integrity and thus avoids the translocation of damaging microbial products [50]. This effect contributes to the hepato-protective properties of the polyphenol at the lowest dose. Moreover, a reduction in the abundance of *Clostridum* sensu stricto 1, whose link with NAFLD-related parameters has been reported elsewhere [51], was also observed in this group. This effect may well be driven by the antioxidant effect of pterostilbene, as reductions in *Clostridum* sensu stricto 1 abundance in response to antioxidant treatment (tempol) have been reported in a murine model of hepatic steatosis [52]. Remarkably, this effect was not found in the rats receiving the high pterostilbene dose (PT30 group), although the hepatic amelioration found in these animals was greater than that found in the PT15 group. Therefore, it could be hypothesised that the hepato-protective effect induced by pterostilbene administration at the lower dose is mediated, at least partially, by changes produced in the abundance of specific gut bacteria. In the case of the animals receiving the high pterostilbene dose (PT30), the present results, along with those previously reported by our group [20,53], suggest that the liver amelioration observed in these animals occurs through other mechanisms. The fact that different doses of pterostilbene induce a variety of changes in the gut microbiota was previously observed in a genetic model of obesity (*fa/fa* Zucker rat) (unpublished data) in our laboratory. In this regard, a biphasic dose–response profile, where low doses exert stimulatory effects and high doses result in inhibitory effects [54,55], has been proposed as an explanation of the efficacy usually found in polyphenols at low doses [56]. This dose–response pattern would rely on the potential role that polyphenols may play as stressors, which induce a defence response in the cells [56]. However, since this hypothesis is not fully confirmed for polyphenols yet, further research is needed to better explain and understand the reasons underlying the observed dose–response differences.

According to the available literature [57], the effect of resveratrol in gut microbiota composition/diversity seems to rely heavily on the animal model and experimental conditions (diet, dose and treatment length). Therefore, the lack of relevant changes found in this precise experimental model could be considered as expectable. Indeed, different authors have suggested that the beneficial effects of resveratrol in gut microbiota mainly occur through changes in intestinal mucosa [58,59,60]. Nevertheless, this is an issue that needs further research.

Although the effects of resveratrol in the amelioration of hepatic steatosis have been extensively investigated, the involvement of the gut microbiota in the observed effects has only been assessed in some of them, and the experimental conditions used in these studies deferred from the ones used in the present work. For instance, in those studies where resveratrol was administered, the diet-induced hepatic damage reached simple steatosis rather than steatohepatitis. Further variations include the use of Kunming mice, C57BL/6J mice or Sprague Dawley rats instead of Wistar rats; the length of the experimental period (6–16 weeks); the use of high-fat or high-fat high-sucrose diets as opposed to a high-fat high-fructose diet; the administration of resveratrol doses of 50–200 mg/kg bw/day; and the resveratrol administration method being oral gavage or dilution in drinking water instead of inclusion in the powdered diet [15,22,60,61].

As far as pterostilbene is concerned, despite the fact that the available data are scarcer, the studies carried out to date have analysed the hepato-protective effect of the compound [62] and the modulation induced in the gut microbiota [23] independently. Thus, to the best of our knowledge, this is the first study that analyses simultaneously the effects of pterostilbene in hepatic steatohepatitis and the modulation induced by this compound in the gut microbiota. On the other hand, it is also important to emphasise that this is the first time that the effects of resveratrol and its dimethoxy derivative pterostilbene on the gut microbiota have been addressed under the same experimental conditions. This represents an important methodological condition to draw comparisons between these two related phenolic compounds.

The present study has some limitations: (a) the method used for gut microbiota analysis (16SrRNA) allows analysis at the genus level, but no further microbe genes that may be related to the metabolism of other compounds of interest have been identified, and (b) the inability to determine whether the observed outcomes regarding pterostilbene and resveratrol administration were driven by the original intact compounds or caused by the metabolites produced in the gut.

In summary, this study demonstrates that the dramatic changes induced in the gut microbiota by high-fat high-fructose feeding are barely ameliorated by pterostilbene and resveratrol. Moreover, under the experimental conditions described in this study, the specific changes in microbiota composition induced by pterostilbene supplementation at the tested doses (increased abundances of *Akkermansia* and *Erysipelatoclostridium,* and lowered abundance of *Clostridum* sensu stricto 1) may not entirely explain the putative preventive effects on steatohepatitis. A similar conclusion may well be drawn for resveratrol, whose effects regarding microbiota modulation were more limited than those of pterostilbene. Nevertheless, as a result of the limitations of this study, further research is needed to better understand the implication of the gut microbiota in the beneficial effects exerted by the tested compounds.

## Figures and Tables

**Figure 1 nutrients-13-01738-f001:**
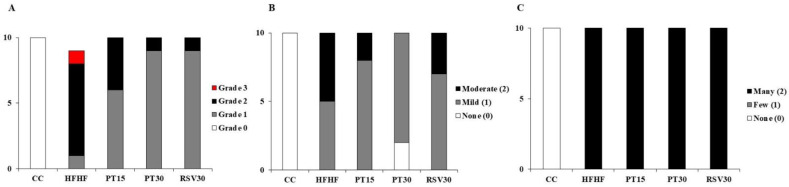
Number of rats showing no steatosis, grade 1, 2 or 3 steatosis (**A**), none, mild or moderate inflammation (**B**) and none, few or many areas of ballooning (**C**) in rats fed with a control diet (CC), a high-fat high-fructose diet (HFHF), a high-fat-high-fructose diet and pterostilbene at a dose of 15 mg/kg/d (PT15), a high-fat high-fructose diet and pterostilbene at a dose of 30 mg/kg/d (PT30) or a high-fat high-fructose diet and resveratrol at a dose of 30 mg/kg/d (RSV30). In the HFHF group, steatosis was analysed in 9 animals.

**Figure 2 nutrients-13-01738-f002:**
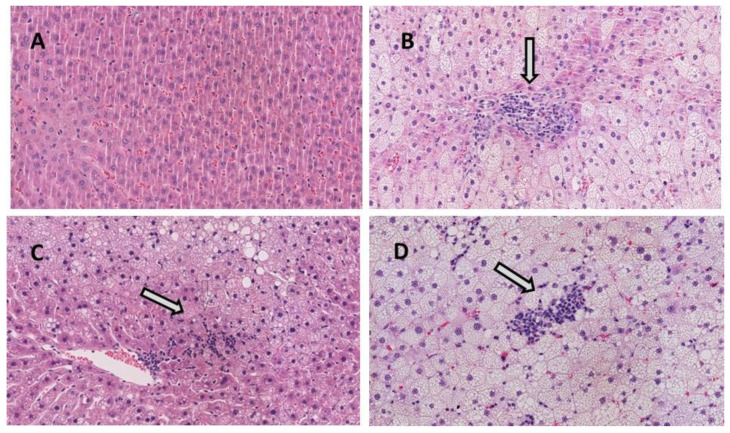
Representative H&E-stained histological liver samples (magnification ×20). (**A**) Healthy liver from CC group, (**B**) liver from HFHF group showing moderate inflammation and ballooning degeneration, (**C**) liver from PT30 group showing mild inflammation and ballooning degeneration and (**D**) liver from RSV30 group showing mild–moderate inflammation and ballooning degeneration. White arrows indicate inflammation. Ballooned cells are identified by the larger size of their wispy cleared cytoplasm.

**Figure 3 nutrients-13-01738-f003:**
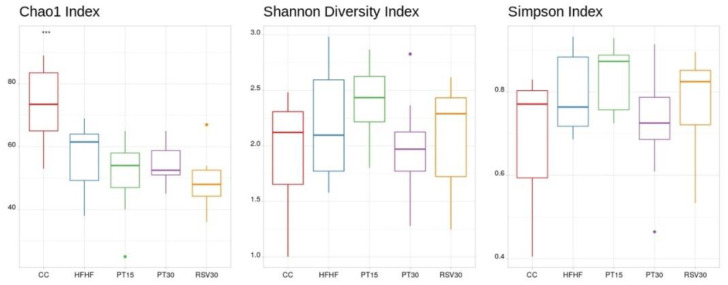
Microbial diversity of the studied groups according to Chao1, Shannon and Simpson. *** *p* < 0.001. Statistical analysis performed using Kruskal–Wallis test and Wilcoxon test.

**Figure 4 nutrients-13-01738-f004:**
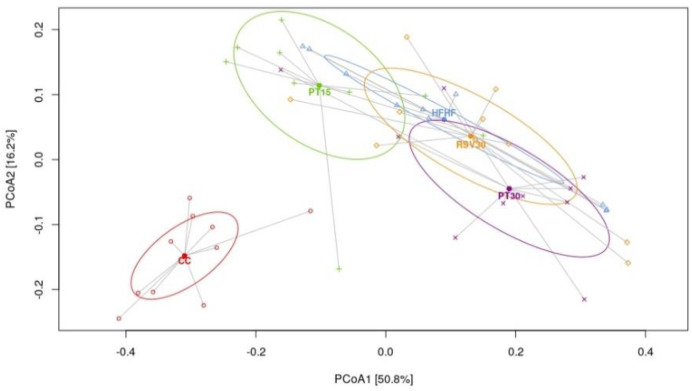
PCoA plot Weighted Unifrac plot. Components PCoA1 and PCoA2 are shown. All samples are connected to the centroid (shown as a point) of the treatment or control group to which they belong.

**Figure 5 nutrients-13-01738-f005:**
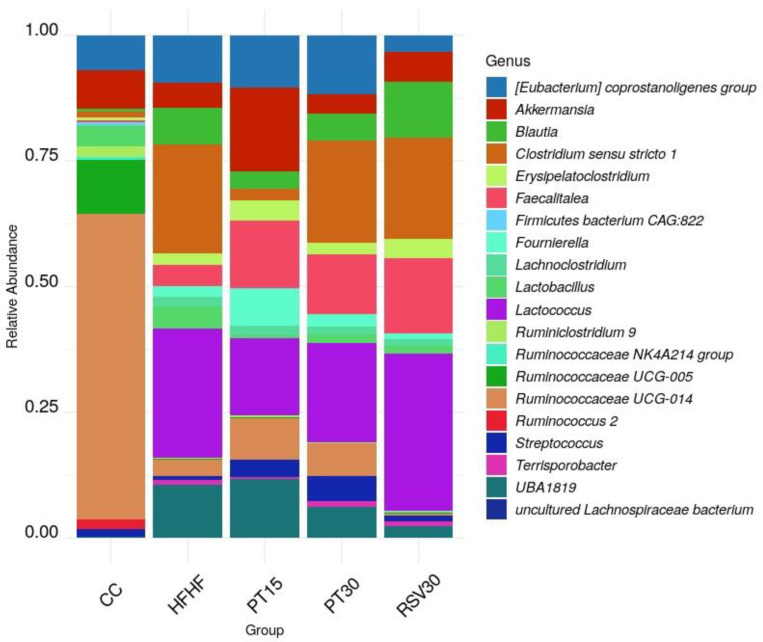
Histogram representing the twenty most abundant genera.

**Figure 6 nutrients-13-01738-f006:**
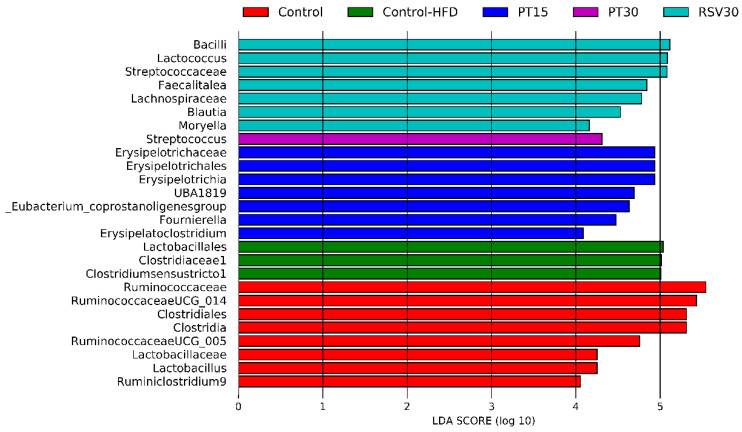
Linear discriminant analysis (LDA) integrated with effect size (LEfSe). Differentially abundant taxonomic groups the microbiota of rats after treatment with pterostilbene and resveratrol (α = 0.05, LDA score > 4).

**Table 1 nutrients-13-01738-t001:** Body weight increase, food efficiency, liver weight and serum transaminase levels of rats fed with the experimental diets for 8 weeks.

	CC	HFHF	PT15	PT30	RSV30	ANOVA
BW increase (%)	88 ± 5 ^b^	108 ± 5 ^a^	109 ± 4 ^a^	106 ± 6 ^a^	93 ± 8 ^ab^	*p* < 0.05
Food efficiency (ΔBW/kcal)	0.041 ± 0.001 ^b^	0.044 ± 0.002 ^ab^	0.046 ± 0.001 ^a^	0.042 ± 0.002 ^ab^	0.040 ± 0.002 ^b^	*p* < 0.05
Liver weight (g)	10 ± 1 ^c^	19 ± 1 ^a^	19 ± 1 ^a^	18 ± 1 ^ab^	16 ± 1 ^bc^	*p* < 0.05
ALT (μmol/min)	27 ± 4 ^c^	229 ± 86 ^a^	125 ± 21 ^b^	103 ± 11 ^b^	120 ± 20 ^b^	*p* < 0.05
AST (μmol/min)	53 ± 3 ^b^	114 ± 24 ^a^	70 ± 6 ^b^	71 ± 8 ^b^	81 ± 7 ^b^	*p* < 0.05

Values are means ± SEM of rats (*n* = 10) fed with a control diet (CC), a high-fat-high-fructose diet (HFHF), a high-fat-high-fructose diet supplemented with 15 mg/kg/d pterostilbene (PT15), a high-fat-high-fructose diet supplemented with 30 mg/kg/d pterostilbene (PT30) or a high-fat-high-fructose diet supplemented with 30 mg/kg/d resveratrol (RSV30) for 8 weeks. Values in the same row with different letters are significantly different at *p <* 0.05, as determined by Newman–Keuls test. ALT: alanine aminotransferase, AST: aspartate aminotransferase, BW: body weight.

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
