# Peer review of "Gut Microbiota Induced by Pterostilbene and Resveratrol in High-Fat-High-Fructose Fed Rats: Putative Role in Steatohepatitis Onset"

_nutrients, 2021, doi:10.3390/nu13051738_

Round 1
Reviewer 1 Report
In this manuscript, Milton-Laskibar et al. investigated that effect of pterostilbene and resveratrol on gut microbiota and permeability in high-fat high-fructose diet fed rat. What is significant about this study is it shows that even under conditions where pterostilbene and resveratrol administration alleviates hepatic steatosis and lobular inflammation, no dramatic changes in gut microbiota occurred. Although the causal relationship between the two cannot be shown from this study alone, the results suggest that the mechanism of action of pterostilbene and resveratrol may be other than changes in gut microbiota. However, data presentation and its quality are inadequate.
Specific Comments:
- The histological study in Figure 1. is not novel, but it is important data to show that the experiment of this study is valid. The figure is very difficult to read and needs to be improved by using colors. In addition, since there should be a histological image that can be used as a criterion for judgment, typical histological images showing the grade of steatosis should be presented, and what is defined as inflammation foci and what is defined as ballooning should be specifically indicated.
- The only data directly related to the barrier function of the intestine is the Western blotting in Figure 6, and it is difficult to discuss the improvement of barrier function by pterostilbene from this data alone. It is necessary to remove this figure and leave the discussion of barrier function for another paper, or to add another data showing the improvement of barrier function. In addition, this Western blotting image is very poor. If this figure is not removed, a better resolution image showing the entire membrane should be included.
- There are some typos, such as “Ruminococcaceae UGC-014” and “Clostridium senso stricto 1”. The manuscript should be carefully checked.
- (C) is missing from the legend in Figure 1.
- In line 308, “no changes” had better to be replaced by "no significant changes".
Author Response
Referee: 1
Comments to the Author
The histological study in Figure 1. is not novel, but it is important data to show that the experiment of this study is valid. The figure is very difficult to read and needs to be improved by using colors. In addition, since there should be a histological image that can be used as a criterion for judgment, typical histological images showing the grade of steatosis should be presented, and what is defined as inflammation foci and what is defined as ballooning should be specifically indicated.
- Following the reviewer's suggestion, Figure 1 has been modified to better differentiate the bars representing the different experimental groups.
- As far as histological images are concerned, they have been included in this revised version, as suggested by the reviewer (Figure 1.2).
The only data directly related to the barrier function of the intestine is the Western blotting in Figure 6, and it is difficult to discuss the improvement of barrier function by pterostilbene from this data alone. It is necessary to remove this figure and leave the discussion of barrier function for another paper, or to add another data showing the improvement of barrier function. In addition, this Western blotting image is very poor. If this figure is not removed, a better resolution image showing the entire membrane should be included.
- According to the reviewer's comment, data concerning proteins involved in intestinal barrier function has been removed from the revised version of the manuscript.
There are some typos, such as “Ruminococcaceae UGC-014” and “Clostridium senso stricto 1”. The manuscript should be carefully checked.
- The reviewer is right. We have revised the manuscript and typos have been corrected.
(C) is missing from the legend in Figure 1.
- The reviewer is right. The “C” letter has been included in the legend of figure 1 in the revised version of the manuscript.
In line 308, “no changes” had better to be replaced by "no significant changes".
- The reviewer is right, and the suggested change has been included in the revised version of the manuscript.
Reviewer 2 Report
The manuscript "Gut microbiota and permeability changes induced by pterostilbene and resveratrol in high-fat-high-fructose fed rats: putative role in steatohepatitis onset" is well-written, clear and easy to read. The introduction provides relevant information, the methods are carefully described and the results are clearly described although the quality of the figures supporting the results obtained must be improved in quality. Figures 3, 4, 5, S1 and S2 would be much more clear and informative if colors are used.
Specific comments:
Introduction
page 1, line 42: ref 3 does not mention fat accumulation grater than 5%. Ref 14 does. Please check.
page 1, line 46-47: please clarify that oxidative stress and mitochondrial disfunction contribute to inflammation.
page 2, line 340-341: ref 14 refers the utilization of resveratrol to study its effects on hepatic steatosis along with different feeding conditions. The authors say "such as resveratrol or pterostilbene" which does not match with ref 14. Please add a reference for a study using both phenolic compounds or add a reference of a study were pterostilbene was used for the same propose.
Material and Methods
page 3, lines 115-116: the sentence " After sacrifice..." is a repetition of what had been said above, on the lines 107-108. Later on line 124 it is described a procedure involving a histological study right after the sacrifice meaning that it was performed on a portion of the freshly excised liver. A brief mention to that portion could be included when describing the procedure on the lines 107-108.
page 4, line 162: dada2 in capital
Results
Table 1: adipose tissue and gastrocnemius muscle weights are mentioned but these values are not shown or even described on the manuscript.
page 5, line 224-225: The steatosis grade is described for each rat of the HFHF group, however it totalizes 9 animals out of 10. Although the explanation about the 10th animal appears later on the manuscript, it could be added to the sentence (line 224-225) for better understand the results presented.
page 5, line 226: a space between "4" and "grade " is missing.
page 5, line 238: there are missing spaces between numbers and ± .
Figure 1: It misses (C) after "ballooning" to describe Figure 1C.
A space is missing between the caption of Figure 5 and the text.
Discussion
page 9, line 332-333: The authors refer a "higher bioavailability of pterostilbene". Is this compared to resveratrol bioavailability? Once the amounts of both phenolic compounds (PT30 and RV30) used in the present study are equal, could you please clarify?
References
References article's titles need to be formatted in a consistent way.
Author Response
Comments to the Author
Introduction
page 1, line 42: ref 3 does not mention fat accumulation greater than 5%. Ref 14 does. Please check.
- The reviewer is right. The reference has been modified and the list has been renumbered accordingly in the revised version of the manuscript.
page 1, line 46-47: please clarify that oxidative stress and mitochondrial disfunction contribute to inflammation.
- Following the reviewer's suggestion, a clarifying sentence, as well as a reference, have been included addressing the relationship between the referred events and liver inflammation in the revised version of the manuscript.
page 2, line 340-341: ref 14 refers the utilization of resveratrol to study its effects on hepatic steatosis along with different feeding conditions. The authors say "such as resveratrol or pterostilbene" which does not match with ref 14. Please add a reference for a study using both phenolic compounds or add a reference of a study were pterostilbene was used for the same propose.
- The reviewer is right. A reference (#21) addressing the usage of pterostilbene for liver steatosis treatment has been included in the revised version of the manuscript (line 315).
page 3, lines 115-116: the sentence " After sacrifice..." is a repetition of what had been said above, on the lines 107-108. Later on line 124 it is described a procedure involving a histological study right after the sacrifice meaning that it was performed on a portion of the freshly excised liver. A brief mention to that portion could be included when describing the procedure on the lines 107-108.
- The reviewer is right. This part of the articles has been modified according to the reviewers´comments.
page 4, line 162: dada2 in capital
- The change requested by the reviewer has been included in the revised version of the manuscript.
Table 1: adipose tissue and gastrocnemius muscle weights are mentioned but these values are not shown or even described on the manuscript.
- The authors appreciate the reviewer's comment. These data were decided not to be included in the manuscript, although mistakenly, its mention in the title of table 1 was not removed. This error has been corrected in the revised version of the manuscript.
page 5, line 224-225: The steatosis grade is described for each rat of the HFHF group, however it totalizes 9 animals out of 10. Although the explanation about the 10th animal appears later on the manuscript, it could be added to the sentence (line 224-225) for better understand the results presented.
- Following the reviewer's suggestion a mention regarding the number of animals used in the HFHF group to analyse steatosis grade has been included in line 224-225 in the revised version of the manuscript.
page 5, line 226: a space between "4" and "grade " is missing.
- The reviewer is right. A space has been included between "4" and "grade ".
page 5, line 238: there are missing spaces between numbers and ± .
- The reviewer is right. The requested spacing has been added in the revised version of the manuscript.
Figure 1: It misses (C) after "ballooning" to describe Figure 1C.
- The reviewer is right. The “C” letter has been included in the legend of figure 1 in the revised version of the manuscript.
A space is missing between the caption of Figure 5 and the text.
- The reviewer is right. The requested spacing has been added in the revised version of the manuscript.
page 9, line 332-333: The authors refer a "higher bioavailability of pterostilbene". Is this compared to resveratrol bioavailability? Once the amounts of both phenolic compounds (PT30 and RV30) used in the present study are equal, could you please clarify?
- Yes, the comment in pterostilbene higher bioavailability is compared to that of resveratrol. Note that since we didn´t check such differences in bioavailability between resveratrol and pterostilbene, this part of the manuscript was a hypothesis. The text has been modified, according to the reviewer's comment.
- What we wanted to emphasize in the referred sentence is that when given at the same dose (30 mg/kg bw/d), pterostilbene not only was as effective as resveratrol in preventing high-fat high-fructose diet induced steatosis, but that it was more effective than resveratrol against inflammation induced by the diet. Therefore, this hepatic anti-inflammatory effect, that was significantly lower in resveratrol, could well be attributed to the higher bioavailability of pterostilbene compared to resveratrol. Nevertheless, as we also mention in this same paragraph, this effect could also be attributed to the chemical structure of pterostilbene.
References
References article's titles need to be formatted in a consistent way.
- The reference format has been homogenized.
Round 2
Reviewer 1 Report
The authors have responded to all of my comment appropriately.